# Modifying the Secretome of Mesenchymal Stem Cells Prolongs the Regenerative Treatment Window for Encephalopathy of Prematurity

**DOI:** 10.3390/ijms25126494

**Published:** 2024-06-12

**Authors:** Josine E. G. Vaes, Suzanne M. Onstwedder, Chloe Trayford, Eva Gubbins, Mirjam Maas, Sabine H. van Rijt, Cora H. Nijboer

**Affiliations:** 1Department for Developmental Origins of Disease, University Medical Center Utrecht Brain Center and Wilhelmina Children’s Hospital, Utrecht University, 3508 AB Utrecht, The Netherlands; 2Department of Neonatology, University Medical Center Utrecht Brain Center and Wilhelmina Children’s Hospital, Utrecht University, 3508 AB Utrecht, The Netherlands; 3Department of Instructive Biomaterials Engineering, MERLN Institute for Technology-Inspired Regenerative Medicine, Maastricht University, 6229 ER Maastricht, The Netherlands

**Keywords:** preterm birth, Encephalopathy of Prematurity, oligodendrocytes, central nervous system, mesenchymal stem cells, glia, experimental models, cellular therapy, cell migration, adenovirus

## Abstract

Clinical treatment options to combat Encephalopathy of Prematurity (EoP) are still lacking. We, and others, have proposed (intranasal) mesenchymal stem cells (MSCs) as a potent therapeutic strategy to boost white matter repair in the injured preterm brain. Using a double-hit mouse model of diffuse white matter injury, we previously showed that the efficacy of MSC treatment was time dependent, with a significant decrease in functional and histological improvements after the postponement of cell administration. In this follow-up study, we aimed to investigate the mechanisms underlying this loss of therapeutic efficacy. Additionally, we optimized the regenerative potential of MSCs by means of genetic engineering with the transient hypersecretion of beneficial factors, in order to prolong the treatment window. Though the cerebral expression of known chemoattractants was stable over time, the migration of MSCs to the injured brain was partially impaired. Moreover, using a primary oligodendrocyte (OL) culture, we showed that the rescue of injured OLs was reduced after delayed MSC coculture. Cocultures of modified MSCs, hypersecreting IGF1, LIF, IL11, or IL10, with primary microglia and OLs, revealed a superior treatment efficacy over naïve MSCs. Additionally, we showed that the delayed intranasal administration of IGF1-, LIF-, or IL11-hypersecreting MSCs, improved myelination and the functional outcome in EoP mice. In conclusion, the impaired migration and regenerative capacity of intranasally applied MSCs likely underlie the observed loss of efficacy after delayed treatment. The intranasal administration of IGF1-, LIF-, or IL11-hypersecreting MSCs, is a promising optimization strategy to prolong the window for effective MSC treatment in preterm infants with EoP.

## 1. Introduction

Encephalopathy of Prematurity (EoP) is a major cause of neurological morbidity in (extreme) preterm neonates [1]. In these infants, white matter development is particularly impacted, characterized by widespread hypomyelination in the absence of cystic lesions (diffuse white matter injury, dWMI) [2,3]. An arrest in oligodendrocyte (OL) lineage maturation, due to preterm birth-related insults, is believed to underlie the observed myelination deficits [4,5]. To date, no clinically approved treatment options to restore dWMI in preterm infants are available.

Preclinical evidence supporting a beneficial role of mesenchymal stem cell (MSC) therapy for dWMI has grown [6]. We, and others, have demonstrated using experimental models that the (intranasal) administration of MSCs after dWMI effectively improves myelination and the functional outcomes, whilst attenuating neuroinflammation [7,8,9]. Using a range of neonatal brain injury models, we and others have shown that transplanted MSCs are unlikely to integrate into brain parenchyma but rather modulate their secretome, contributing to a cerebral environment permissive for repair and neurogenesis through paracrine signaling [10,11,12]. Most recently, we showed that intranasal MSC therapy potently restored myelination after early administration (i.e., 3 days (D3) after dWMI) in newborn mice [9]. However, the therapeutic potential of intranasal MSCs decreased significantly when treatment was postponed until day 6 (D6) [9]. A narrow treatment window could potentially limit the clinical applicability of intranasal MSC therapy in extreme preterm infants, as the early identification of dWMI is challenging due to its multiple-hit pathophysiology, first MRI possibility, and a lack of reliable biomarkers [3,13,14]. Here, we hypothesized that the reduced efficacy of postponed intranasal MSC treatment could be the result of either impaired MSC homing at a later treatment timepoint, or to a limited regenerative potential of MSCs in later stages of dWMI pathophysiology. In the first case, a possible lack of chemotactic factors crucial for MSC homing to areas of the dWMI at later timepoints could be responsible. In the latter case, the optimization of the MSC secretome, i.e., boosting their trophic and anti-inflammatory properties, could potentially prolong the treatment window.

As a follow-up to our previous work [9], we assessed cerebral chemotactic signals between D3 and D6 after dWMI using ex vivo PCR arrays. Furthermore, we studied MSC homing using nanoparticle-based cell tracing. We used primary glial cultures to investigate the potential superior capacity of MSCs overexpressing insulin-like growth factor 1 (IGF1), the epidermal growth factor (EGF), the leukemia inhibitory factor (LIF), interleukin 10 (IL10), or interleukin 11 (IL11), secreted factors previously identified as beneficial for OL maturation or the dampening of microglia activation in vitro [9]. Finally, we explored whether the intranasal administration of IGF1-, IL11-, LIF-, or IL10-overexpressing MSCs could prolong the treatment window for dWMI in our mouse model.

## 2. Results

### 2.1. Intranasal MSC Treatment Efficacy in dWMI: Timing Matters

In our recent study, we showed that the combination of postnatal inflammation and hypoxia/ischemia in P5 mice induced a pattern of brain injury that closely mimics preterm dWMI. In this model, we observed that the intranasal application of MSCs restores dWMI on both a functional and anatomical level when MSCs are applied relatively early (i.e., D3) after the insult. In the current study, we confirm the limited treatment window for intranasal MSCs, when administration was delayed until D6 after the dWMI. Figure 1A,B shows that the dWMI-induced reduction of cortical myelination (*p* = 0.0075, compared to sham) at P26 was potently restored after MSC treatment at D3 (*p* = 0.027, compared to vehicle treatment). However, when treatment was postponed until D6, the MSCs failed to restore the cortical myelination. In our previous study, similar conclusions were drawn after an assessment of the myelin microstructure (Figure 1C).

### 2.2. Expression of Chemotactic Signals in the Brain following dWMI Induction

To assess which chemotactic factors may be involved in MSC migration to the brain, the cerebral gene expression profiles of dWMI mice versus the sham control mice at D3 were analyzed. We identified the differential expression of six chemokines (Ccl4, Cxcl10, Ccl3, Cxcl3, Cxcl5, and Cxcl1) associated with the migration of MSCs or other cell types in the literature (Table 1) [15,16,17]. Subsequently, we compared the expression of these cerebral chemokines in dWMI mice sacrificed at D6 versus D3. The expression of Cxcl10 and Cxcl1 were (further) upregulated at D6 compared to D3 after injury. The expression of Ccl4, Ccl3, Cxcl3, and Cxcl5 remained as high at D6 compared to D3 after injury (Table 1). The expression of two factors (i.e., Cxcl10 and Ccl3) were confirmed by real-time RT-PCR using the individual samples and showed similar fold regulation changes compared to the arrays (Appendix A). Our data indicate that the loss of efficacy at D6 of the MSC treatment is probably not primarily caused by a lack of chemotactic signals in the brain at D6 compared to D3.

### 2.3. MSCs Change their Secretome In Situ after Treatment Delay

To investigate changes in the paracrine functioning of MSCs after delayed administration, we analyzed the gene expression profiles of MSCs exposed to brain extracts of dWMI mice at D6 versus D3 in vitro, using PCR arrays. We identified a difference in 42 MSC-expressed factors after exposure to the cerebral milieu at D6 versus D3 (Table 2). The expression of trophic factors secreted by MSCs after D6 vs. D3 brain extract exposure remained unchanged [9]. Interestingly, D6 brain extract exposure resulted in upregulation of pro-inflammatory cytokines and Bmps, including Tnf, IL1β, IL2, and Bmp2 in MSCs (Table 2). The expression of two factors (i.e., IL1β and Ccl3) were confirmed by real-time RT-PCR using the individual samples (Appendix A).

### 2.4. Treatment Delay Limits MSC Migration after Intranasal Administration

MSCs labeled with mesoporous silica-coated gold nanoparticles were used to study migration following intranasal administration at D3 and D6 after dWMI. The distribution of cells was measured by the detection of gold signals in the tissue homogenates, using inductively coupled plasma mass spectrometry (ICP-MS). The postponement of treatment to D6 tended to reduce (~50%) the total amount of gold found in the injured brain at 12 h after administration, compared to D3 treatment (*p* = 0.065) (Figure 2A). In line with our previous findings at D3, the majority of gold was detected in the brain following intranasal delivery, with minimal loss in the liver, lungs, or spleen (*p* = 0.075, *p* = 0.074, *p* = 0.035 brain compared to liver, lungs and spleen, respectively) (Figure 2B). Moreover, we observed the dispersed distribution of cells throughout the diffusely injured brain (Figure 2C).

### 2.5. MSC Modification Leads to Hypersecretion of Selected Factors

To optimize the MSCs’ secretome, we genetically engineered MSCs to transiently overexpress one selected factor previously identified as beneficial for OL maturation and/or dampening of microglia activation, namely IGF1, EGF, LIF, IL11, and IL10 [6,9]. The MSCs were transduced at different multiplicity of infections (MOIs) (Table 3). The effect of the transduction on selected factor secretion was measured using ELISA. We observed a 17×, 10×, 5×, 39×, and 9× increase in the mean IGF1, EGF, IL11, LIF, and IL10 concentrations, respectively, at the optimal MOI, when compared to the mean concentrations secreted by EV-MSCs (in bold, Table 3). The successful adenoviral vector infection of MSCs was confirmed visually, using an eGFP signal (Appendix A).

### 2.6. Modification of the MSC Secretome Enhances Myelination and Prolongs the Treatment Window In Vitro

To study the possible superiority of modified MSCs to boost OL maturation and subsequent myelin production, we cultured primary pre-OLs and challenged these with a medium of LPS-stimulated microglia (microglia-conditioned medium (MCM)+LPS) in a non-contact coculture with MSCs. A 24 h treatment interval (i.e., adding the MSC transwell inserts 24 h after MCM+LPS) was used to mimic the delay to the MSC treatment in vivo. Pre-OLs exposed to MCM+LPS demonstrated a strong reduction in the MBP+ area compared to pre-OLs exposed to MCM−LPS (dotted line) (*p* = 0.002), indicating impaired maturation (Figure 3A,B). The direct coculture with empty vector (EV) MSCs partially restored the MBP+ area in pre-OL cultures exposed to MCM+LPS (*p* = 0.002) (Figure 3A). The beneficial effect of EV-MSCs was identical to that of naïve, non-modified MSCs, as observed in our previous study (Figure 3C) [9]. IGF1, LIF, or IL11-MSCs demonstrated a superior treatment efficacy in the MBP+ area, when compared to EV-MSCs (*p* = 0.015, *p* = 0.0089, and *p* = 0.024 vs. EV-MSCs, respectively) (Figure 3A,B). The coculture with EGF-MSCs or IL10-MSCs did significantly boost myelin production compared to empty gel inserts (*p*= 0.0082 and *p* = 0.041 vs. empty gels, respectively), but failed to significantly outperform the EV-MSCs (*p* = 0.063 and *p* = 0.275 vs. EV-MSCs, respectively) (Figure 3A,B). When the start of the coculture was delayed for 24 h, the EV-MSCs were not able to improve the myelin production by OLs after MCM+LPS. Furthermore, we observed an overall reduction in the efficacy of the modified MSCs, though IGF1 and IL10-MSCs were still able to significantly improve myelination compared to the empty gel inserts (IGF1 *p* = 0.044 and IL10 *p* = 0.020) and to the EV-MSCs (IL10-MSCs *p* = 0.032 and a trend for IGF1-MSCs (*p* = 0.066) (Figure 3D).

### 2.7. Modified MSCs Display Superior Anti-Inflammatory Properties on Microglia In Vitro

To assess the direct effects of the secretome modification of MSCs on microglia activation, we exposed primary LPS-stimulated microglia to a non-contact coculture with modified MSCs. LPS stimulation strongly increased Tnfα production by microglia compared to non-stimulated cells (*p* < 0.0001) (Figure 3E). A coculture with EV-MSCs significantly decreased Tnfα production, indicating attenuation of neuroinflammation (*p* = 0.011). The secretome of EGF, LIF, and IL10-MSCs displayed a superior dampening effect on Tnfα production by microglia (*p* = 0.0007, *p* = 0.0006 and *p* = 0.0004, respectively, versus EV-MSCs) (Figure 3E). A coculture with IGF1 or IL11-MSCs did not significantly reduce Tnfα secretion to a superior level in regard to EV-MSCs (*p* = 0.4932 and *p* = 0.4537, respectively; *p* = 0.0008 and *p* = 0.0011, respectively, versus empty inserts).

To study the environmental changes provoked by the coculture with modified MSCs, we measured the concentration of 31 different cytokines and chemokines in the microglia supernatants, using Luminex. We observed distinct and specific micro-environmental changes in 13 factors when comparing the exposure of microglia to EV-MSCs with the different types of modified MSCs (Appendix A, in bold).

### 2.8. Intranasal Administration of Modified MSCs Prolongs the Treatment Window after dWMI

To investigate the therapeutic potential of modified MSCs after treatment delay, dWMI animals received modified MSCs intranasally at D6. Based on the in vitro findings on OL maturation, we selected IGF1-, LIF-, IL11-, and IL10-overexpressing MSCs as the most promising candidates to prolong the treatment window. The complexity of the myelin microstructure was assessed using segmentation analyses at P26, as described before [18]. The EV-MSC treatment at D6 failed to significantly restore the dWMI-induced reduction in fiber length and the number of intersections (*p* = 0.228 and *p* = 0.168, respectively), indicating persistent myelination failure after delayed MSC treatment, as we observed before (Figure 4A,B, [9]). Interestingly, treatment with IGF1, LIF, or IL11-MSCs at D6 significantly improved the fiber length and number of intersections (IGF1-MSC: *p* = 0.024 and *p* = 0.003, IL11-MSC: *p* = 0.041 and *p* = 0.047, LIF-MSC: *p* = 0.011 and *p* = 0.0003, versus vehicle) (Figure 4A,B). The intranasal administration of IL10-MSCs did not significantly improve the myelin microstructure (*p* = 0.908 and *p* = 0.483, versus vehicle).

In line with the histological findings and our earlier study [9], intranasal EV-MSCs at D6 failed to significantly improve the motor outcome at P26 (*p* = 0.7778) (Figure 4D). Treatment with IGF1, IL11, or LIF-MSCs at D6 potently reduced the forepaw preference (*p* = 0.0002, *p* = 0.0024, and *p* = 0.0013, respectively, versus vehicle) (Figure 4D). Similar to our histological findings, IL10-MSCs did not improve the motor performance (*p* > 0.999 versus vehicle). Taken together, these data indicate a superior therapeutic efficacy of IGF1, IL11, and LIF-MSCs on dWMI after delayed treatment onset.

### 2.9. Modified MSCs Attenuate Neuroinflammation after Delayed Administration following dWMI

In line with the previous findings in this model, we observed an increase in the number of Iba+ cells in the corpus callosum of vehicle-treated dWMI animals compared to sham-control animals at P26 (*p* = 0.033) [9]. The treatment with EV or modified MSCs at D6 did not significantly reduce the Iba+ numbers, though we observed a trend with IL11-MSCs (*p* = 0.08) (Figure 5A,B). More detailed analyses of the microglial morphology revealed an amoeboid (activated) phenotype in the vehicle-treated dWMI animals, displayed by an increase in the cell circularity and solidity (*p* < 0.0001 and *p* < 0.0001 vs. sham control, respectively). The administration of IL10 and LIF-MSCs reduced cell circularity and solidity (circularity: *p* = 0.012 and *p* = 0.041 and solidity: *p* = 0.005 and *p* = 0.026, respectively), while other modified MSCs or EV-MSCs did not significantly reduce the activation state of the microglia (Figure 5C,D). The astrocyte reactivity was assessed in the corpus callosum and hippocampus at P26. Similar to our previous study [9], the induction of dWMI led to an increase in the GFAP+ area in the brains of vehicle-treated dWMI animals compared to sham-control animals (corpus callosum *p* = 0.004 and hippocampus *p* < 0.0001). The administration of IGF1-MSCs dampened the astrocyte reactivity in both the corpus callosum and the hippocampus (*p* = 0.028 and *p* = 0.025, respectively) (Figure 6A–D). In addition, treatment with IL11 and IL10-MSCs reduced the astrocyte activation in the hippocampus (*p* = 0.015 and *p* = 0.034, respectively), but not in the corpus callosum (Figure 6A–D).

## 3. Discussion

In this study, we investigated the potential superior capacity of genetically modified MSCs (i.e., hypersecreting IGF1, EGF, LIF, IL10, or IL11) to prolong the treatment window for dWMI in newborn mice after intranasal application. We confirm, here, that the treatment window of intranasal MSCs in our mouse model of dWMI is limited, with a strong reduction in treatment efficacy when MSCs are administered at D6 versus D3 after dWMI. Though the cerebral chemotactic signals after dWMI appear to remain largely intact between D3 and D6, we show that the migration of MSCs after intranasal administration is hampered at D6. Furthermore, we show that naïve MSCs exposed to D6 brain extracts ex vivo respond with similar secreted growth factor profiles, but a more pro-inflammatory profile when compared to MSCs exposed to D3 brain extracts. Moreover, in vitro assays, using primary OL cultures, reveal a limited potential of naïve MSCs to boost myelination after delayed coculture. Taken together, these results indicate that both impaired cell homing, as well as a limited regenerative potential of naïve MSCs in the later stages of dWMI pathophysiology could underlie the observed loss of treatment efficacy. To optimize the treatment window of intranasal MSC treatment, MSCs were successfully modified to transiently overexpress IGF1, EGF, IL11, LIF, or IL10. We report, here, a superior capacity of selected modified MSCs to directly boost OL maturation and attenuate microglia activation in vitro, with unique environmental changes provoked by the different modified MSC types. Moreover, we show that the intranasal administration of IGF1, LIF, or IL11-MSCs restores myelination and improves the behavioral outcome when applied at D6 after dWMI. In addition, LIF and IL10-MSCs dampen microglia activation after D6 treatment. Furthermore, IGF1, IL11, and IL10-MSCs reduce astrocyte reactivity after administration at D6. Collectively, these data imply that modified MSC treatment is a potent strategy to prolong the treatment window in preterm dWMI, using cells with a superior regenerative potential to compensate for impaired cell migration and enduring injury.

A broad therapeutic window is essential for the clinical translation of novel treatments for preterm infants, as pinpointing the exact timeframe in which dWMI develops is challenging. The pathophysiology of dWMI is believed to be multifactorial, with multiple (potentially) detrimental insults occurring in the perinatal and (early) postnatal period [2,4]. Currently, clinical diagnosis of dWMI is often based on neuro-imaging around term-equivalent age, when myelination is progressing [19]. Moreover, validated biomarkers for the early identification of preterm neonates at risk of the developmental of brain injury are lacking. Thus, while the early administration of MSCs could be vital for optimal treatment efficacy, selecting patients that may possibly benefit from MSC therapy could be difficult in an early phase. Therefore, the prolongation of the therapeutic window by using modified MSCs might prove to be very relevant for this group of patients. Previous studies in the field of (neonatal) brain injury have reported a superior treatment efficacy of genetically engineered MSCs that (transiently) hypersecrete a beneficial factor versus naïve MSCs [20,21,22]. However, genetic engineering of cells is often met with some safety concerns. The adenoviruses used here do not integrate into the MSCs’ DNA and, thus, induce only transient overexpression of the gene of interest [23,24]. Moreover, it is believed that intranasally administered MSCs are short lived and do not integrate into the brain, but temporarily aid endogenous repair through paracrine signaling [25]. Pioneering clinical studies, using modified human MSCs in adult stroke patients did not report safety concerns [26]. Additional preclinical studies assessing the long-term outcomes and conventional clinical safety studies are needed to confirm safety in regard to preterm neonates.

Chemokine gradients in tissues are crucial to regulate the migration of MSCs to sites of injury [27]. Moreover, in order for MSCs to exert their regenerative capacities, close proximity to the lesion site is suggested to be of importance [11,28,29]. Here, we identified changes in expression of six chemokines associated with chemotaxis of MSCs or other cells at D3 after dWMI [15,16,17]. When comparing the cerebral expression profiles of these chemokines at D6 versus D3, we observed an even further upregulation of Cxcl10 and Cxcl1 at D6, while the expression of Ccl3, Ccl4, Cxcl3, and Cxcl5 remained unchanged. Importantly, murine MSCs have been reported previously to express the receptors for most of these ligands, i.e., CCR3, CCR5, and CXCR3 [30]. However, the expression of CXCR2 was reported to be low in murine MSCs, implying a limited role for Cxcl1 in MSC chemotaxis [30]. Even though cerebral chemokine levels were stabile between D3 and D6, gold-labeled MSC tracing experiments did reveal a borderline significant reduction in the amount of MSCs reaching the brain at D6 versus D3 after dWMI. The observed discrepancy between the cerebral chemokine levels and cell tracing might be explained by several things. In the acute phase of perinatal brain injury, the integrity of the blood–brain barrier (BBB) is reportedly comprised [31,32]. Though intranasally administered MSCs have been shown to be able to bypass the BBB [33], an intact BBB in the later stages of dWMI could potentially play a role in impaired cell migration. Moreover, it is possible that changes in other chemoattractants important for MSC homing, e.g., the basic fibroblast growth factor (bFGF), vascular endothelial growth factor (VEGF), hepatocyte growth factor (HGF), insulin-like growth factor 1 (IGF1), platelet-derived growth factor (PDGF) and osteopontin, which were not included in our PCR array, play a role in impairing MSC migration [27,34,35]. Aside from limitations in the chemotactic factors assessed in the array, it is possible that factors that did not reach our cut-off in up- or downregulation could have an essential role in MSC migration after dWMI. From the current data, it is unclear if MSCs truly migrate to a lesser extent after intranasal administration at D6, leading to a smaller proportion of cells responsible for regeneration, or if the migration process is slowed down. In the latter case, the observed reduction in treatment efficacy at D6 might be explained by impaired regenerative capacity of an equal amount of MSCs arriving in the later stages of dWMI pathophysiology.

As suggested above, an impaired regenerative potential of MSCs in the later stages of OL injury could underlie the observed reduction in the therapeutic efficacy after delayed MSC treatment. Previous studies in other (neonatal) brain pathologies report similar findings, with MSC therapy being most effective during acute inflammation and a reduction in efficacy after disease stabilization [36,37]. We, and others, have shown that MSCs adapt their secretome based on the microenvironment of the target tissue, which indicates that insufficient endogenous production of inflammatory cytokines in situ at later timepoints after dWMI might not elicit essential secretome changes in MSCs [9,10,36]. In line with this hypothesis, we observed changes in the secretome of MSCs exposed to the D6 intracerebral milieu compared to D3, with a largely unchanged expression of trophic factors, but an upregulation of known OL differentiation-inhibiting factors, such as Tnfα, Bmp2, IL1b, IL2, IL17, and IL9 [4,38].

To boost the potential of MSCs for delayed treatment, we modified the secretome. In vitro, we show that IGF1, LIF, and IL11-MSCs were superior in boosting OL maturation compared to control EV-MSCs. Similar to our in vivo findings, EV-MSCs did not significantly improve the OL maturation after in vitro delay of the MSC coculture. Interestingly, IL10 and IGF1-MSCs were able to (borderline) significantly boost OL maturation compared to control EV-MSCs after an initial delay in the coculture. Apart from affecting OL maturation, we assessed the impact of modified MSCs on microglia activation, another key pathophysiological hallmark of dWMI. EGF, LIF, and IL10-MSCs were shown to outperform EV-MSCs in attenuating microglial activation in vitro. Based on the combined in vitro findings, we selected IGF1, IL11, LIF, and IL10-MSCs for in vivo administration at D6. EGF-MSCs were excluded, as these cells did not outperform EV-MSCs in boosting OL maturation at either time points in the culture. IGF1, IL11, and LIF-MSCs were able to significantly improve myelination and the functional outcome after administration at D6, thereby prolonging the treatment window. In addition, IGF1-MSCs were shown to dampen the astrocyte reactivity at D6. IL11 and IL10-MSCs selectively reduced astrocyte activation in the hippocampus, but not the corpus callosum. IL10-MSCs failed to significantly improve myelination and motor performance, however these cells were able to attenuate microglia activation at D6. These data imply that the attenuation of microglia activation plays a less prominent role in dWMI repair in the later stages of injury. However, considering the effects of IL10-MSCs on OL maturation in vitro, it is also possible that the secreted levels of IL10 after modification of the MSCs are insufficient for repair at later time points.

Our current data indicate a promising potential for modified MSC treatment in the (delayed) repair of dWMI. However, the exact mechanisms underlying the superior treatment efficacy after postponed treatment remain unclear. The investigation of the microglial supernatant after coculture with different hypersecreting MSCs revealed a specific composition of factors present in the culture medium per MSC type, implying that different modified MSCs could elicit unique environmental changes in situ. It is unclear if these changes are solely evoked by selective hypersecretion of the factor of interest secreted by MSCs. The observed net equal regenerative repair response after modified MSC treatment at D6 compared to repair by naïve MSCs at D3 could be the result of a superior, high dose of the overexpressed trophic factor as such, or of a superior cocktail of (multiple) secreted beneficial factors evoked by autocrine actions, leading to increased OL maturation though a smaller amount of MSCs reaching the lesion at D6. Additional studies are needed to explore whether the overexpression of individual trophic factors lead to changes in the rest of the modified MSCs’ secretome. Aside from a superior regenerative capacity, improved migration of modified MSCs could also play a role in the observed therapeutic efficacy after delayed treatment. Multiple strategies for MSC priming (for example with hypoxic or inflammatory stimuli) and modification have been shown to influence cell migration with an upregulation of receptors vital for MSC migration [27,35].

## 4. Materials and Methods

All procedures were carried out according to the Dutch and European guidelines (Directive 86/609, ETS 123, Annex II) and were approved by the Experimental Animal Committee Utrecht (Utrecht University, Utrecht, The Netherlands) and the Central Authority for Scientific Procedures on Animals (the Hague, The Netherlands) (project identification code: AVD115002016751, date of approval: 1 December 2016). Detailed materials and methods can be found in the Appendix A.

### 4.1. MSC Culture

GIBCO^®^ mouse (C57BL/6) bone marrow-derived MSCs (Invitrogen, S1502-100; Carlsbad, CA, USA) were cultured in D-MEM/F-12 medium with 10% fetal bovine serum (10565-018 and 12662-029, Invitrogen), according to the supplier’s protocol. The MSCs were passed once (from P2 to P3) prior to in vivo administration or in in vitro experiments.

### 4.2. MSC Transfection

The MSCs were modified to transiently overexpress growth factors or cytokines with ready-to-use recombinant adenoviral vectors encoding a murine IGF1, EGF, LIF, IL11, or IL10 transgene, combined with a control eGFP vector to assess infection efficacy (Vector Biolabs, Malvern, PA, USA). The MSCs were plated 24 h before infection, with 2.0 × 10^5^ cells per well in 6-well plates, followed by exposure to viral particles for 6 h. Thereafter, the cells were recultured for 24 h, followed by in vitro gel embedment, or in vivo administration. The optimal multiplicity of infection (MOI) was determined as per the adenovirus, through the assessment of the secreted protein using ELISA at 2 days after infection.

### 4.3. In Vivo Model of Diffuse White Matter Injury

Specifically, dWMI was induced at postnatal day 5 (P5) in C57BL/6j mouse pups, as described previously [9]. Hyaluronidase was administered to the nasal cavity 30 min prior to administration of 0.5 × 10^6^ MSCs [9]. The MSCs were administered at P8 (i.e., D3) or P11 (i.e., D6). Previous dose-response experiments identified 0.5 × 10^6^ MSCs as the lowest effective dose [9]. Vehicle-treated dWMI animals received dPBS. The mice were euthanized at P8, P11, or P26 (i.e., 3 weeks) by an i.p. pentobarbital overdose. For a schematic overview of the model, we refer to Figure 7. For the PCR arrays, the sham control or dWMI (untreated) brains were collected, the cerebellum was discarded, and the hemispheres were separately snap frozen in liquid nitrogen, and stored at −80 °C until further processing.

#### 4.3.1. Cerebral Chemokine Expression Profiles

The sham control and dWMI (untreated) brains were collected at D3 and D6 after dWMI. The RNA was isolated from the ipsilateral hemispheres and cDNA transcription was carried out. The cDNA of the sham control or dWMI animals was pooled per time point (D3 n = 5 and D6 n = 4 per experimental condition) and a PCR array (PAMM-150Z, Qiagen, Hilden, Germany) was performed. The chemokine/cytokine gene expression changes were calculated: (1) in dWMI mice versus sham-control mice at D3 to identify chemokines that were differentially regulated following injury, and (2) in dWMI mice at D6 versus D3 to study the stability of the chemotactic signals over time. A fold regulation threshold of 3.0 was considered as either down- or upregulation. The PCR array results were validated by quantitative PCR analyses in the individual cDNA samples for selected genes (Appendix A). The primer sequences can be found in Appendix A. The mean expression of GADPH and β-actin were used for data normalization.

#### 4.3.2. MSC Gene Expression Profiles after Exposure to Brain Extracts

The brains were collected at D3 (n = 5) and D6 (n = 4) and brain extracts were made. The MSCs were cultured and seeded at 2.0 × 10^5^ cells per well. After 24 h, the culture medium was replaced with knock-out DMEM containing either the D3 or D6 brain extract at a concentration of 1 mg protein/mL. After 48 h, the MSC RNA was isolated and transcribed to cDNA and the PCR arrays were performed (Qiagen; PAMM-041Z and PAMM-150Z). The gene expression changes in the MSCs exposed to D6 dWMI brain extracts were calculated relative to the D3 dWMI brain-extract exposure. A fold regulation threshold of 3.0 was considered as either down- or upregulation. The PCR array results were validated for selected genes (Appendix A). The primer sequences can be found in Appendix A.

### 4.4. MSC Labeling and Cell Tracing

The MSCs were labelled using gold core-mesoporous and lipid-coated silica nanoparticles (AuNP-MSN-LIP). A detailed description of the nanoparticle synthesis, characterization, and labeling efficiency can be found in our previous paper [9]. In short, 2 h after cell passaging, the MSCs were incubated with 25 µg/mL AuNP-MSN-LIP in a culture medium over 48 h. Following cell labeling, the dWMI animals received intranasally 0.5 × 10^6^ MSCs at D3 or at D6. Then, 12 h after treatment, the mice were sacrificed by a pentobarbital overdose, the brains were dissected and frozen in liquid nitrogen, as well as the spleen, lung, and liver. For details on inductively coupled plasma mass spectrometry (ICP-MS) to quantitatively assess MSC biodistribution through the detection of gold in mouse tissue homogenates, we refer to [9] and the Appendix A.

### 4.5. Immunohistochemistry

At P26, the animals were sacrificed by a pentobarbital overdose, followed by transcardial perfusion with PBS and 4% PFA. The brains were post-fixed for 24 h in 4% PFA, followed by dehydration in ethanol. The brains were paraffin-embedded and coronal sections (8 µm) were cut at the hippocampal level (−1.80 mm from the bregma in adult mice). For 3,3′-Diaminobenzidine (DAB) staining, the sections were deparaffinized and rehydrated, blocked in 20% normal rabbit serum (NRS) in PBS/0.1% Tween and incubated overnight with rat-anti-MBP (MAB386, Merck Millipore; 1:500, Burlington, MA, USA) in 10% NRS/PBS/0.1% Tween. For visualization, we used biotinylated rabbit-anti-rat (BA-4000, Vector laboratories, 1:400, Newark, CA, USA) with a vectastain ABC kit (Vector laboratories) and 0.5 mg/mL DAB (Sigma, Burlington, MA, USA), followed by embedment in depex (Serva, Heidelberg, Germany). For immunofluorescent staining, sections were deparaffinized and rehydrated, heated to 95 °C in sodium citrate buffer (0.01 M, pH 6) for antigen retrieval, blocked with 10% normal goat serum in PBS + 0.1% Tween20 for MBP/NF200 or 2% bovine serum albumin (BSA)/0.1% saponin in PBS for Iba1/GFAP staining, followed by overnight incubation with rat-anti-MBP (MAB386, Merck Millipore; 1:500), rabbit-anti-NF200 (N-4142, Sigma; 1:400), rabbit-anti-Iba1 (019-19741, Wako; 1:500, Neuss, Germany), and with mouse-anti-GFAP (BM2278, Origine; 1:200, Herford, Germany). Subsequently, the sections were incubated with alexafluor-594 and -488 conjugated secondary antibodies (Life technologies, Carlsbad, CA, USA; 1:200–500), followed by DAPI (1:5000) counterstaining and embedment in Fluorsave (Merck Millipore, 345789). In between steps, we used PBS as a washing buffer.

### 4.6. Microscopy and Image Analysis

The investigators were blinded to the experimental conditions during image acquisition and analysis. In the MBP-DAB-stained sections, 2.5× magnification was used to create an image of the ipsilateral hemisphere using a light microscope (Zeiss, Oberkochen, Germany), with an AxioCam ICc 5 camera (Zeiss). For the immunofluorescent staining, a Cell Observer microscope with an AxioCam MRm camera (Zeiss, Oberkochen, Germany) was used to acquire images of the ipsilateral hemisphere. For the MBP/NF200 staining, 3 adjacent 40× micrographs were taken at a fixed distance from the external capsule into the cortex (for exact locations see [18]). For the Iba1/GFAP staining, two 20× images were acquired in the corpus callosum of the ipsilateral hemisphere. Moreover, a 20× image of the CA1 region of the hippocampus was made via GFAP staining.

For both the cortical myelination (2.5×) on MBP-DAB staining and the microstructural integrity of the myelinated axons (40×) using the MBP/NF200 stained sections, we refer to [18]. The morphology of microglia residing in the corpus callosum was assessed via skeleton analyses (circularity and solidity), after manual selection using the particle analysis function in ImageJ v.1.47 software [39], as described by [40]. GFAP threshold analyses were carried out using ImageJ software v.1.47 to determine the GFAP+ area of the staining in the corpus callosum and hippocampus. The values of all the acquired images were averaged per animal.

### 4.7. Behavioral Assessment

The motor performance was evaluated using the cylinder rearing test (CRT) at P26, as described in [9] and the Appendix A. In our previous study, the CRT was shown to be a reliable test to evaluate the functional outcome in our model [9]. In short, the animals were placed in a transparent cylinder. The forepaw preference was calculated as ((non-impaired − impaired)/(non-impaired + impaired + both)) × 100%. All the CRTs were videotaped and scored by researchers blinded to the experimental conditions.

### 4.8. In Vitro Models of dWMI

#### 4.8.1. Primary Rat Glial Cultures

A mixed glial culture was acquired from P1-2 Sprague Dawley rat pup cortices, as described by [41], with small changes made by our group [9]. To mimic the in vivo inflammatory situation to induce maturation arrest in immature oligodendrocytes, microglia were plated at a cell density of 0.5 × 10^6^ cells per well in poly-L-ornithine (Sigma Aldrich, Burlington, MA, USA, BurP3655)-coated 24-well plates and MCM was produced, as described previously. OPCs were isolated and plated at 4.0 × 10^4^ cells/well on poly-D,L-ornithine (Sigma Aldrich, P0421)-coated 24-well plates for the OL differentiation experiments.

#### 4.8.2. Primary Mouse Microglia Culture

A primary microglia culture was prepared from P1 C57BL/6 mice cortices for the cocultures, as described previously [9]. After isolation, microglia were seeded in poly-L-ornithine-coated 24-well plates at a density of 1.5 × 10^5^ cells per well. The cocultures (see below) were started 24 h later.

#### 4.8.3. Non-Contact MSC Glia Cocultures

At 24 h prior to the start of the cocultures (and 24 h after MSC transfection) 4.0 × 10^4^ modified MSCs (MSC eGFP (control, empty vector (EV) MSC), MSC IGF1, MSC EGF, MSC LIF, MSC IL10, and MSC IL11) were embedded in Hydromatrix gel (Sigma, A6982) transwell inserts (Merck Millipore, MCHT24H48), according to the supplier’s protocol.

For the OL differentiation experiment, the OPC medium containing pro-proliferation factors (see Appendix A) was replaced with either MCM+LPS or MCM−LPS when the majority of OLs displayed an immature pre-OL morphology (i.e., 4 days after OPC plating). Pro-differentiation factors (see Appendix A) were added to MCM+LPS or MCM−LPS to start the differentiation of the OPCs. Transwell inserts containing modified MSCs, with EV-MSCs serving as a negative control, or no MSCs as an empty insert control, were added to the wells directly or 24 h (delayed) after the induction of differentiation. The inserts were removed 72 h after the addition of MCM and the OLs were fixated with 4% PFA in PBS for 10 min.

For the microglia experiment, at 24 h after plating, the cocultures of MSCs and mouse microglia were started by adding 50 ng/mL LPS (Sigma, L4515) and putting the transwell inserts containing modified MSCs into the wells. After 48 h of the coculture, the inserts were removed and the microglia supernatant was collected, aliquoted, and stored at −80 °C for the ELISA.

#### 4.8.4. ELISA

The Tnfα concentrations in the supernatant of microglia were measured using an ELISA kit for murine Tnfα (Ucytech, Utrecht, The Netherlands), according to the manufacturer’s protocol. The Tnfα data of different experiments were normalized to positive control conditions (i.e., 50 ng/mL LPS plus empty insert without MSCs).

#### 4.8.5. Luminex Assay

The concentrations of 31 cytokines/chemokines in the pooled microglia supernatant (n = 3 per condition) were measured using a bioplex pro mouse chemokine assay (12009159, Biorad), according to the supplier’s protocol. All the concentrations were normalized to the EV-MSC condition (i.e., 50 ng/mL LPS plus an insert with EV-MSCs).

#### 4.8.6. Immunocytochemistry of Primary Oligodendrocyte Cultures

After fixation, nonspecific binding was blocked using 2% BSA and 0.1% saponin in PBS, followed by overnight incubation with primary antibodies (rabbit-anti-Olig2, AB9610. Merck Millipore; 1:1000, mouse-anti-MBP, SMI-94, Biolegend; 1:1000, San Diego, CA, USA). Subsequently, the wells were incubated with alexafluor-594 and -488 conjugated secondary antibodies (Life Technologies; 1:1000), followed by Hoechst 33342 (Sigma) for nuclear counterstaining and embedment in Fluorsave (Merck Millipore, 345789). In between the steps, PBS was used as a washing buffer.

Six adjacent fields were imaged (10×), starting at a fixed distance from the well edges. The number of Olig2- and Hoechst- positive cells were counted using the analyze particles function in ImageJ v. 1.47. The area of MBP+ staining was measured using manual thresholding analyses in ImageJ. The thresholds were kept consistent per batch. To compare independent experiments, all the results were normalized for the positive control (MCM+LPS; empty insert without MSCs).

### 4.9. Statistics

All data are shown as the mean ± standard error of the mean (SEM). The statistics were performed using GraphPad Prism 8.3. For details on the statistics, see the Appendix A. Moreover, *p*-values <0.05 were considered statistically significant. The sample sizes are mentioned in the figure captions.

## 5. Conclusions

To summarize, this study shows that the therapeutic window of intranasal MSC therapy in a mouse model of preterm dWMI is relatively limited. The observed reduction in treatment efficacy likely results from the impaired migration of MSCs towards the brain and a limited regenerative capacity in the later stages of dWMI. Modified MSCs, transiently hypersecreting IGF1, IL11, and LIF, possess a superior capacity to boost white matter development after dWMI and, thereby, extend the therapeutic window.

## Figures and Tables

**Figure 1 ijms-25-06494-f001:**
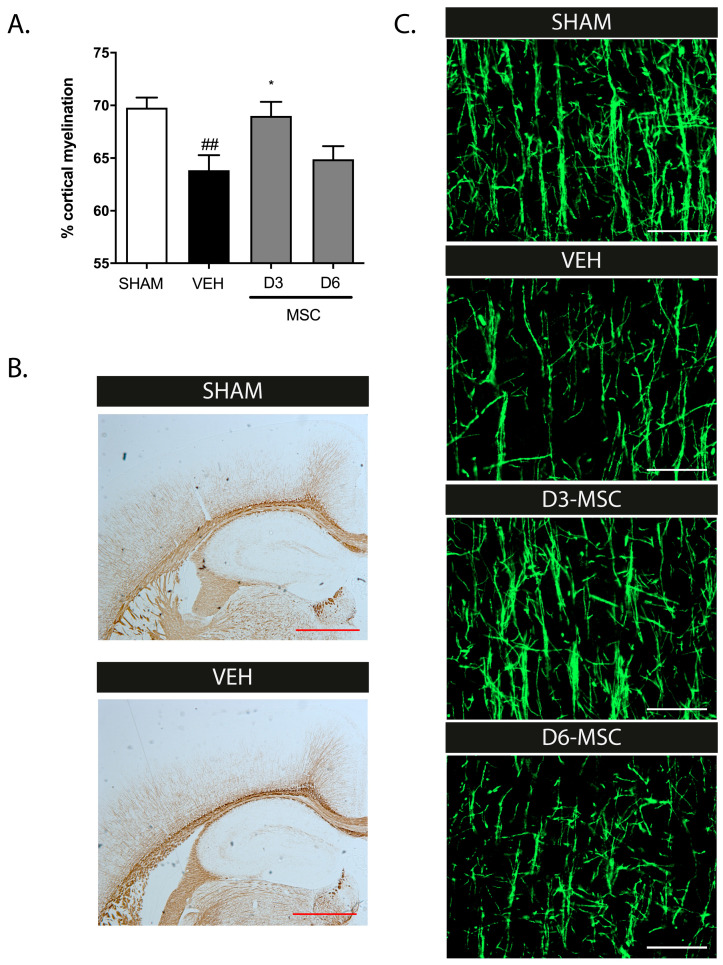
Delayed intranasal MSC administration reduces treatment efficacy. (**A**) Cortical myelination was restored in dWMI animals that received intranasal MSC therapy at D3. Delay in MSC administration to D6 reduced treatment efficacy (SHAM n = 9, VEH n = 7, MSC-D3 n = 8, MSC-D6 = 9). (**B**) Representative MBP-DAB staining of the cortex of a sham control (upper) and dWMI (lower) mouse at P26. Scale bars: 200 µm. (**C**) Representative fluorescent images of MBP+ axons in the ipsilateral cortex of a sham-operated control mouse, a dWMI-vehicle treated mouse, and dWMI mice treated with MSCs at D3 and D6 (from top to bottom). Scale bars: 50 µm. ##: *p* < 0.01; vehicle-treated dWMI animals vs. sham controls; *: *p* < 0.05; MSC D3-treated dWMI vs. vehicle-treated dWMI animals.

**Figure 2 ijms-25-06494-f002:**
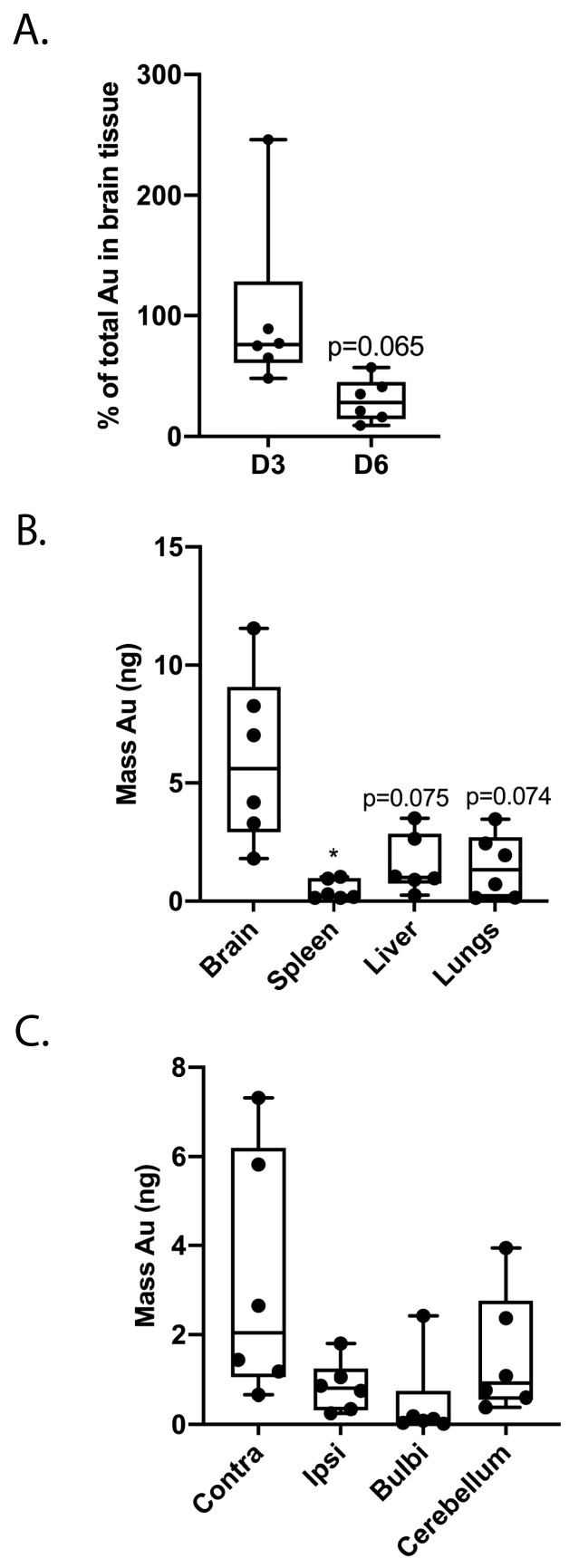
Delayed intranasal MSC treatment is associated with a reduction in the migration of silica-coated gold nanoparticle-labeled MSCs to the brain. (**A**) A reduction (~50%) in the total amount of gold in the injured brain was observed after a delay of the intranasal MSC treatment to D6 vs. D3 (MSC D3 n = 6, MSC D6 n = 6). (**B**) Intranasal MSC administration at D6 is associated with minimal loss of cells in the liver, lungs, or spleen, as the majority of the gold nanoparticles were detected within the brain. (**C**) The amount of gold measured in the brain was evenly distributed throughout the different parts of the injured brain. *: *p* < 0.05; peripheral organs vs. brain. Nearly significant *p* values are indicated in (**A**,**B**).

**Figure 3 ijms-25-06494-f003:**
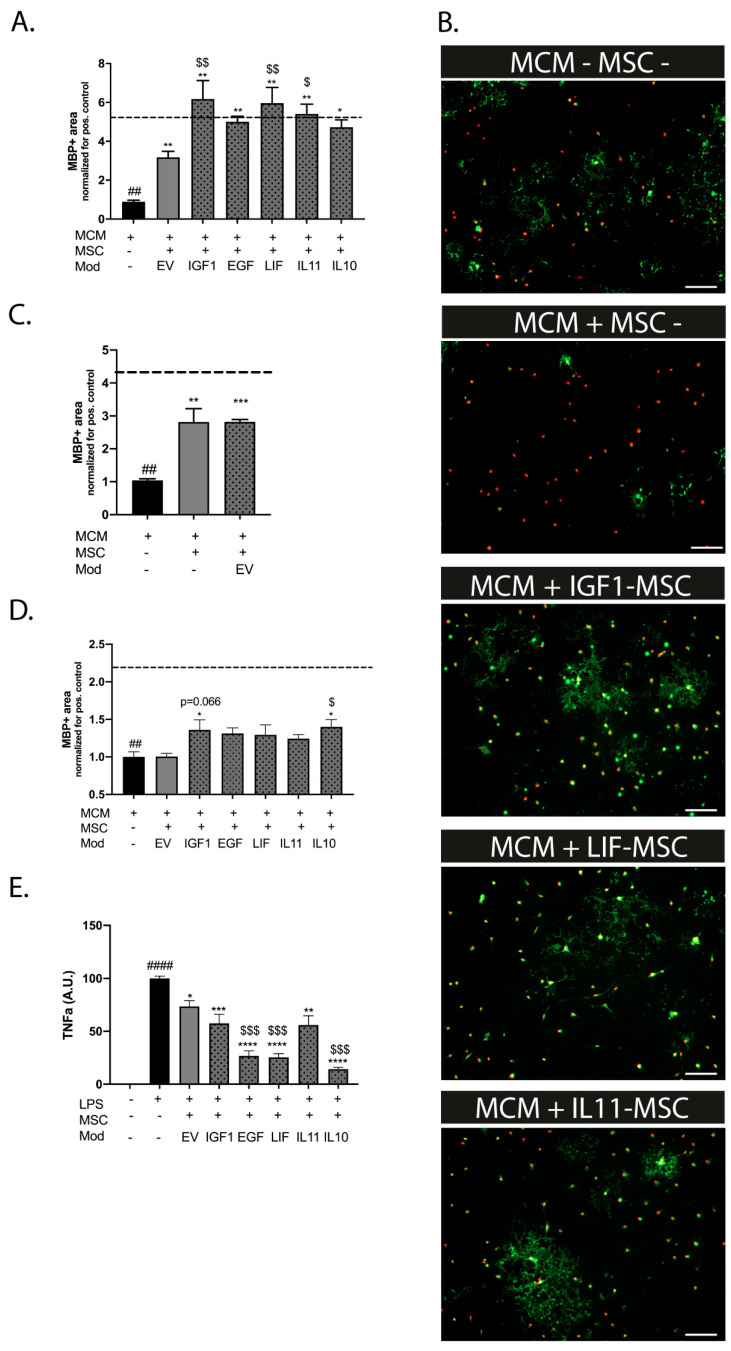
Modification of the MSC secretome enhances myelination in vitro (**A**) MCM+LPS leads to a reduction in the MBP+ area (dashed line represents MBP+ area in MCM−LPS control condition). Treatment with 4 × 10^4^ EV-MSCs, EGF-MSCs, or IL10-MSCs in a non-contact coculture significantly enhances MBP expression. IGF1-MSCs, IL11-MSCs, or LIF-MSCs display a superior ability to boost OL maturation compared to EV-MSCs (n = 3 independent experiments, 3–4 observations per experiment, normalized for the positive control, e.g., cells exposed to MCM+LPS, which was set at 1). Cocultures were started at the same time as MCM exposure. (**B**) Representative fluorescent images of primary cultured oligodendrocytes stained for oligodendrocyte marker Olig2 (red) and myelin component MBP (green). Cells were exposed to MCM−LPS (MCM−) or MCM+LPS (MCM+) in a non-contact gel insert with or without modified MSCs. Scale bars: 100 µm. (**C**) Transfection of MSCs with EV did not affect the capacity to restore the MBP+ area, compared to naïve, non-transfected MSCs (n = 2 independent experiments, 2 observations per experiment, normalized for the positive control, e.g., cells exposed to MCM+LPS, which was set at 1). (**D**) A 24 h delay in the start of the coculture impairs the treatment efficacy of EV-MSCs after in vitro maturation arrest of OLs (MCM+LPS). A coculture with IGF1- and IL10-overexpressing MSCs significantly improves MBP expression, however, with a lower efficacy compared to a direct coculture (see (**A**)). Only IL10-MSCs significantly outperformed EV-MSCs in the restoration of the MBP+ area; IGF1-MSCs had a borderline significant superior effect (n = 2 independent experiments, 3–4 observations per experiment, normalized for the positive control, e.g., cells exposed to MCM+LPS, which was set at 1). (**E**) LPS stimulation evokes a strong increase in Tnfα secretion by microglia. Treatment with 4 × 10^4^ EV-MSCs, IGF1-MSCs, or IL11-MSCs in a non-contact gel insert partially attenuates microglial Tnfα production. A non-contact coculture with EGF-MSCs, LIF-MSCs, or IL10-MSCs leads to additional dampening of Tnfα secretion compared to EV-MSCs (n = 2 independent experiments, 2 observations per experiment, normalized for the positive control, e.g., cells exposed to LPS, which was put at 100). ##: *p* < 0.01; ####: *p* < 0.0001 MCM+ (black bars) vs. MCM− control (dashed line (**A**,**C**,**D**) or versus no LPS in (**E**)); *: *p*< 0.05; **: *p* < 0.01; ***: *p* < 0.001, ****: *p* < 0.0001 MSC conditions vs. MCM+ control (black bar (**A**,**C**,**D**)) or LPS control (black bar (**E**)); $: *p* < 0.05; $$; *p* < 0.01; $$$: *p* < 0.001 modified MSC conditions (gray dotted bars) vs. EV-MSC condition (gray bars). Nearly significant *p* values are indicated in (**D**).

**Figure 4 ijms-25-06494-f004:**
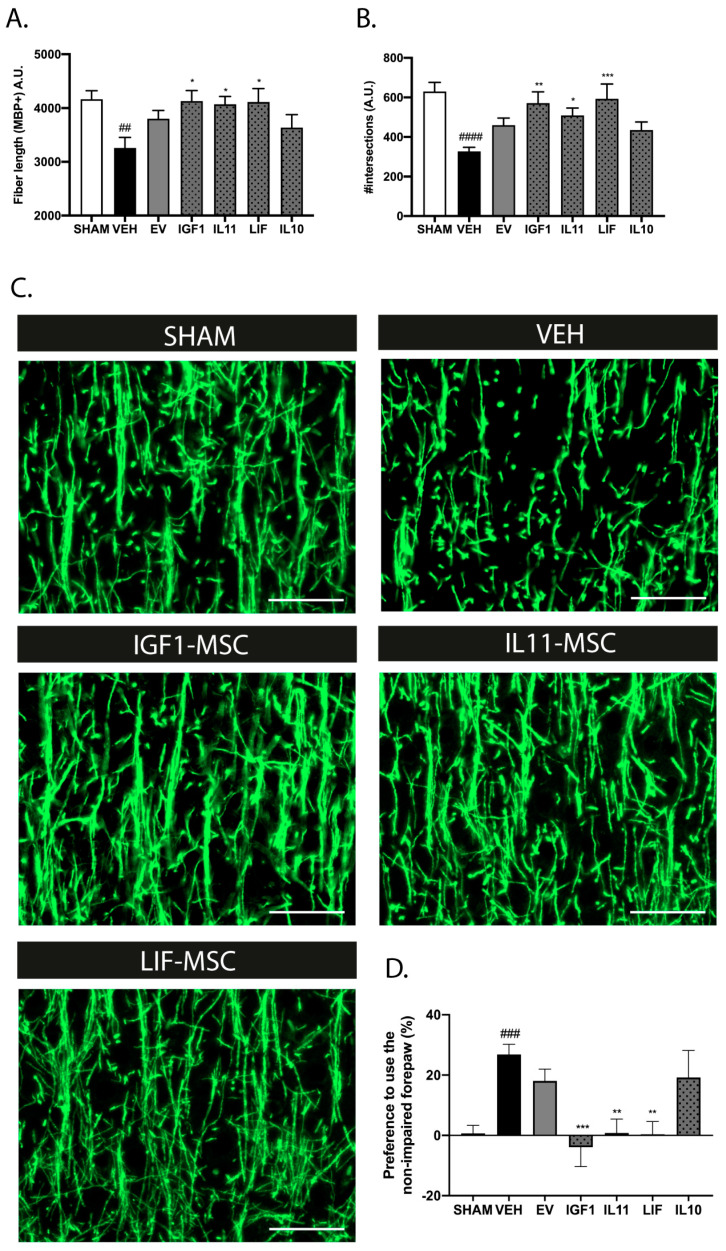
Intranasal administration of modified MSCs prolonged the treatment window in dWMI mice. (**A**,**B**) Intranasal EV-MSC treatment at D6 failed to restore the dWMI-induced reduction in fiber length (**A**) and the number of intersections (**B**), as measures in terms of the myelin microstructure. Treatment with IGF1, LIF, or IL11-MSCs significantly improved the microstructural myelin parameters (SHAM n = 15, VEH n = 15, EV-MSCs n = 13, IGF1-MSCs n = 9, IL11-MSCs n = 9, LIF-MSCs n = 13, and IL10-MSCs n = 11). (**C**) Representative fluorescent images of MBP+ axons in the ipsilateral cortex of a sham-operated control mouse, dWMI-vehicle mouse, and dWMI mice treated with MSC IGF1, MSC IL11, or MSC LIF. Scale bars: 50 µm (**D**) Intranasal administration of EV-MSCs at D6 after dWMI induction failed to reduce forepaw preference in the cylinder rearing test. Intranasally administered IGF1, IL11, or LIF-MSCs significantly improved the motor outcome at D6 (SHAM n = 15, VEH n = 15, EV-MSCs n = 15, IGF1-MSCs n = 10, IL11-MSCs n = 9, LIF-MSCs n = 10, and IL10-MSCs n = 8). ##: *p* < 0.01; ###: *p* < 0.001; ####: *p* < 0.0001 vehicle-treated dWMI animals vs. sham controls; *: *p* < 0.05; **: *p* < 0.01; ***: *p* < 0.001; modified MSC-treated vs. vehicle-treated animals.

**Figure 5 ijms-25-06494-f005:**
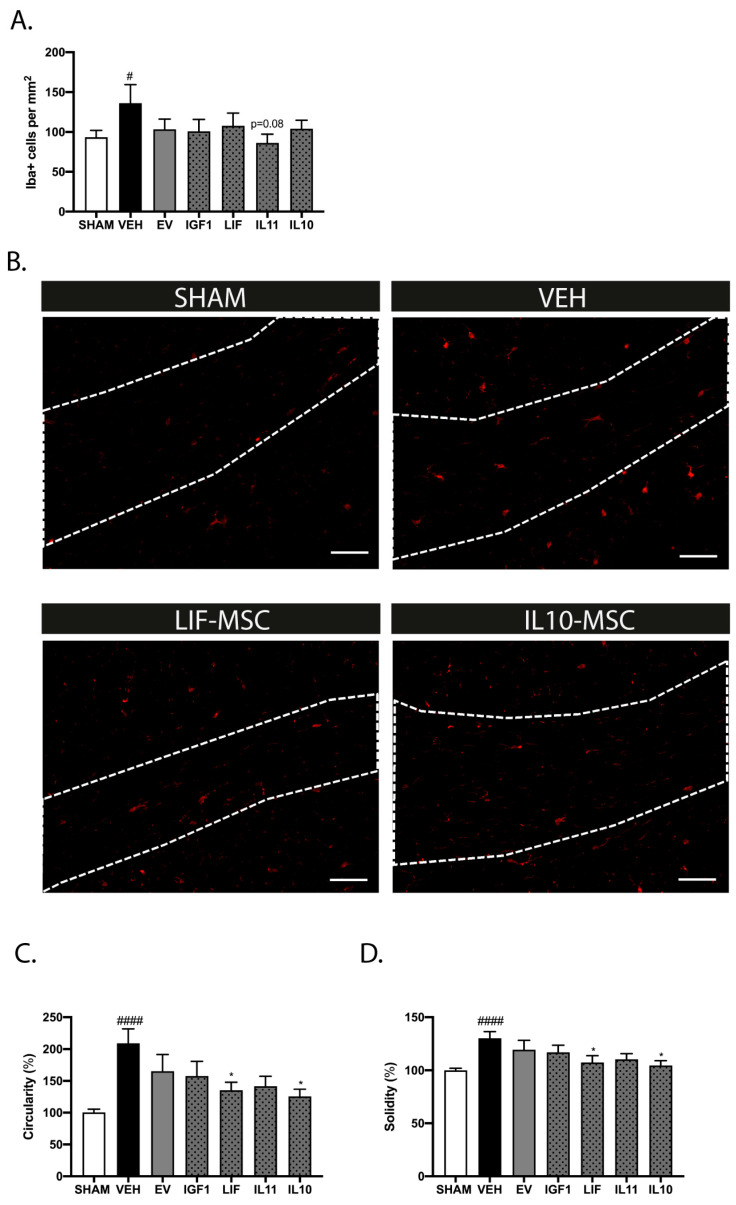
Modified MSCs attenuate microglia activation following delayed administration. (**A**) Quantification of microglia cell numbers in the corpus callosum revealed a reduction in Iba+ cells (trend) after IL11-MSC treatment compared to the vehicle treatment at D6. Treatment with EV-MSCs or other modified MSCs at D6 did not affect the microglia density in the injured brain (SHAM n = 15, VEH n = 14, EV-MSCs n = 13, IGF1-MSCs n = 10, IL11-MSCs n = 9, LIF-MSCs n = 12, and IL10-MSCs n = 10). (**B**) Representative fluorescent images of Iba+ cells in the corpus callosum (white outline) in sham control, vehicle-treated dWMI, LIF-MSC treated, and IL10-MSC-treated dWMI animals. Scale bars: 100 µm. (**C**,**D**) Assessment of microglia circularity (**C**) and solidity (**D**), morphological parameters of the microglia activation state, showed a less pro-inflammatory phenotype following intranasal administration of LIF or IL10-MSCs compared to the vehicle treatment at D6 (SHAM n = 14, VEH n = 12, EV-MSCs n = 9, IGF1-MSCs n = 8, IL11-MSCs n = 9, LIF-MSCs n = 10, and IL10-MSCs n = 11). #: *p* < 0.05; ####: *p* < 0.0001 vehicle-treated dWMI animals vs. sham controls; *: *p* < 0.05; modified MSC-treated vs. vehicle-treated dWMI animals. Nearly significant *p* values are indicated in (**A**).

**Figure 6 ijms-25-06494-f006:**
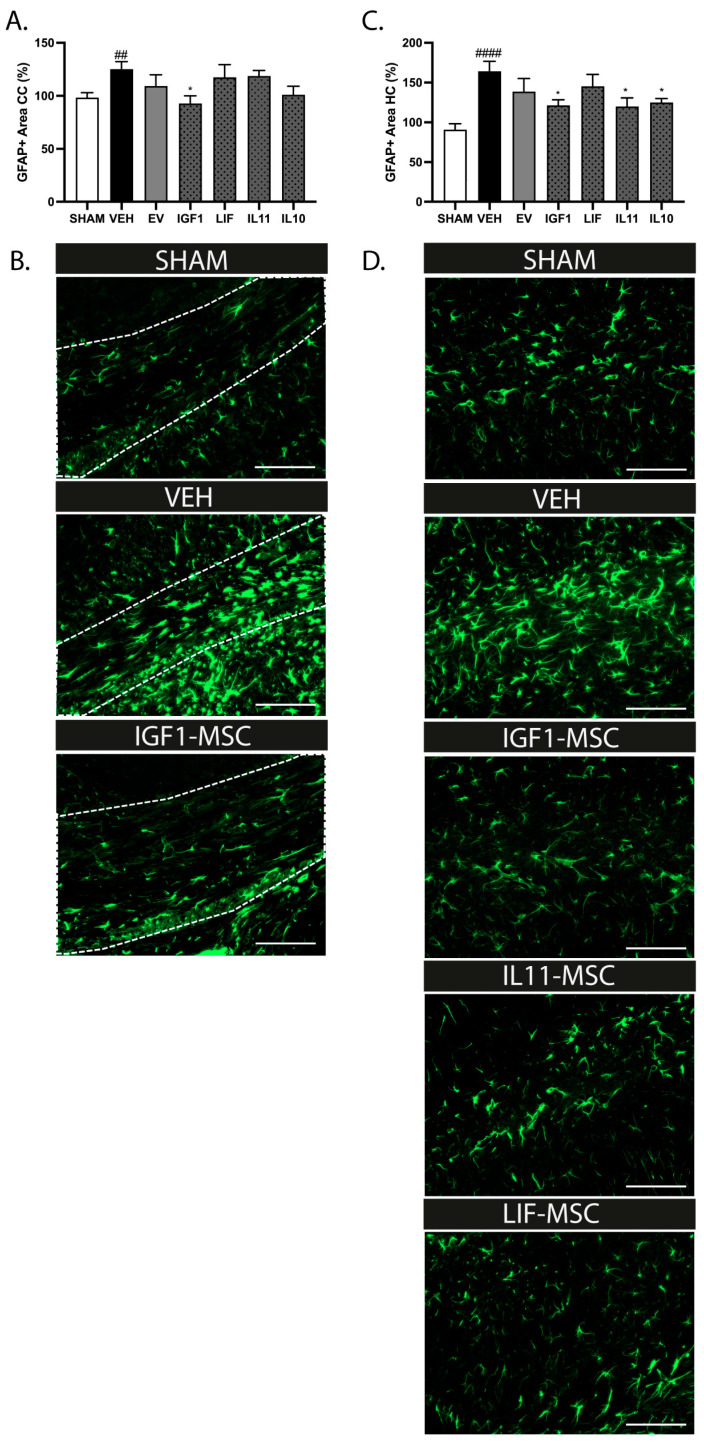
Delayed treatment with modified MSCs at D6 reduced astrocyte reactivity. (**A**) A reduction in the GFAP+ area was observed in the corpus callosum of IGF1-MSC-treated dWMI animals compared to vehicle-treated dWMI animals, as an indication of the reduced astrocyte reactivity (SHAM n = 14, VEH n = 13, EV-MSCs n = 13, IGF1-MSCs n = 9, IL11-MSCs n = 13, LIF-MSCs n = 10, and IL10-MSCs n = 9). (**B**) Representative fluorescent images of GFAP+ cells in the corpus callosum (white outline) in sham control, vehicle-treated dWMI, and IGF1-MSC-treated dWMI animals. Scale bars: 100 µm. (**C**) Delayed administration of IGF1-MSCs, IL11-MSCs, and IL10-MSCs at D6 reduced the GFAP+ area in the hippocampus of dWMI animals compared to vehicle-treated dWMI animals (SHAM n = 14, VEH n = 11, EV-MSCs n = 11, IGF1-MSCs n = 9, IL11-MSCs n = 10, LIF-MSCs n = 9, and IL10-MSCs n = 11). (**D**) Representative fluorescent images of GFAP+ cells in the CA1 region of the hippocampus in sham control, vehicle-treated dWMI, IGF1-MSC-treated dWMI, IL11-MSC-treated dWMI, and IL10-MSC-treated dWMI animals. Scale bars: 100 µm. ##: *p* < 0.01; ####: *p* < 0.0001 vehicle-treated dWMI animals vs. sham controls; *: *p* < 0.05; modified MSC-treated vs. vehicle-treated dWMI animals.

**Figure 7 ijms-25-06494-f007:**
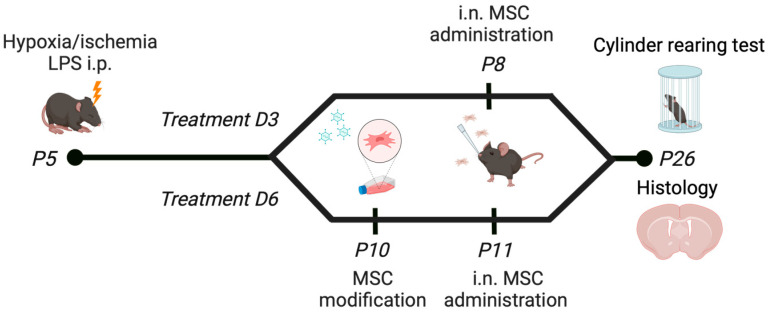
Schematic overview of the in vivo model. Illustration created with BioRender.com. accessed on 23 May 2024.

**Table 1 ijms-25-06494-t001:** Gene expression changes (fold regulation) following dWMI induction.

Symbol	D3 (P8)	dWMI
	dWMI vs. SHAM	D6 (P11) vs. D3 (P8)
Ccl4	9.09	1.22
Cxcl10	6.47	5.21
Ccl3	3.74	−2.86
Cxcl3	3.62	1.51
Cxcl5	3.13	1.02
Cxcl1	−3.57	16.45

**Table 2 ijms-25-06494-t002:** MSC secretome gene expression changes (fold regulation) after treatment delay.

Symbol	D6 vs. D3
Adipoq	3.54
Bmp2	3.20
Bmp7	9.13
Ccl19	5.07
Ccl22	12.58
Ccl24	17.64
Ccl3	4.41
Ccl4	6.04
Csf2	3.42
Cxcl13	12.79
Cxcl3	3.08
Fasl	−6.64
Fgf13	4.72
Ffg3	5.86
Fgf4	3.59
Fgf5	3.61
Fgf8	3.42
Hc	3.07
IfnA2	6.34
IfnG	3.86
Il1b	3.50
Il2	13.65
Il3	3.92
Il4	4.12
Il5	3.04
Il9	14.53
Il10	8.64
Il12b	13.52
Il17a	3.97
Il17f	7.44
Il22	4.33
Il23a	3.33
Il24	6.30
Mstn	7.99
Nodal	7.20
Ntf3	3.94
Osm	309.49
Tnf	7.60
Tnfrsf11b	5.24
Tnfsf10	31.85
Tdgf1	12.51
Xcl1	10.41

**Table 3 ijms-25-06494-t003:** Optimal multiplicity of infection (MOI) for MSC transduction.

Factor	Condition	MOI	Mean Concentration
**IGF1 (pg/mL)**	Control (no virus)	-	30.1
	EV-MSC	2000	23.9
	IGF1-MSC	500	111.9
		1000	352.1
		**2000**	408
**EGF (pg/mL)**	Control (no virus)	-	0
	EV-MSC	4000	0
	EGF-MSC	2000	2.7
		**4000**	9.5
**IL11 (ng/mL)**	Control (no virus)	-	0
	EV-MSC	4000	79.5
	IL11-MSC	2000	231.9
		**4000**	414
**LIF (pg/mL)**	Control (no virus)	-	2.5
	EV-MSC	8000	11.5
	LIF-MSC	4000	278.7
		**8000**	445.6
**IL10 (pg/mL)**	Control (no virus)	-	0
	EV-MSC	4000	420.5
	IL10-MSC	1000	2295
		2000	2634
		**4000**	3725

## Data Availability

The data that support the findings in this study are available from the corresponding author upon reasonable request.

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
