# Peer review of "Modifying the Secretome of Mesenchymal Stem Cells Prolongs the Regenerative Treatment Window for Encephalopathy of Prematurity"

_ijms, 2024, doi:10.3390/ijms25126494_

Round 1

Reviewer 1 Report

Comments and Suggestions for Authors

In their paper, Vaes and colleagues present evidence demonstrating that modifying the secretome of mesenchymal stem cells (MSCs) enhances oligodendrocyte (OL) maturation and reduces microglia activation. This effect was observed both in vitro and in vivo using a double-hit mouse model of diffuse white matter injury. The authors propose that genetic engineering of MSCs represents a promising strategy to extend the window for effective MSC treatment in preterm infants with Encephalopathy of Prematurity (EoP).

However, as noted below, there are areas where the paper could be improved through revisions:

1) Although various protocols have been described in detail in the supplementary methods, the manuscript could be more reader-friendly if some details were reiterated in the results section. For example:

1a) I would suggest briefly describing the premature encephalopathy model in section 2.1 of the results, perhaps with an illustrative figure of the protocol.

1b) To facilitate the reader in the results chapter 2.3, I would suggest mentioning in the first sentence that these are in vitro studies where MSCs were exposed to brain extracts and gene expression was obtained through PCR array.

2) Section 2.5 requires clarification and rephrasing to better align with the section title.

2a) I suggest modifying the sentence "we genetically engineered MSCs to transiently overexpress a selected factor previously identified" to "we genetically engineered MSCs to transiently overexpress one selected factor previously identified at a time" and to explicitly mention the selected factors at this point.

2b) Additionally, I propose representing and discussing the secretome results in the main text, as they are currently only mentioned in a table in the supplementary material for the calculation of the Multiplicity of Infection.

3) The quality of the images in Figure 3B is very low. Please replace the images and add an inset at a higher magnification.

4) Representative images of the presented results are often missing. Please add:

4a ) In figure 3B and 4 representative images of all the conditions (or at least of IGF1-, LIF-, and IL11-MSCs)

4b) In Figure 5, images demonstrating the decrease in microglia circularity and solidity for the statistically significant conditions.

5) Regarding microglial morphological analysis, what about the other parameters measured in the cited paper by Zanier et al. (i.e., area, perimeter, Feret’s diameter (caliper), and aspect ratio)? Are they unchanged?

6) In their previous work (Vaes JEG, van Kammen CM, Trayford C, et al. Intranasal mesenchymal stem cell therapy to boost myelination after encephalopathy of prematurity. Glia. 2021;69:655–680. https://doi.org/10.1002/glia.23919), the authors show that MSC treatment also attenuates astrocyte activation following dWMI. Please check if this effect is boosted by genetically engineering MSCs

Minor:

Line 130: Write out "ICP-Ms" in full.

Line 163: Specify "EV" as "Empty Vector."

Line 212: The alpha symbol for "α" of Tnf appears strange visually.

Line 337: “ […] while expression of Ccl3, Ccl4, Cxcl3 and Cxcl5 remained unchanged”. Accordingly to Table 1 Ccl3 is reduced. If this is the case please modify the sentence.

Line 346: MSCS should be MSCs

Line 349: ie or eg?

Reviewer 2 Report

Comments and Suggestions for Authors

In the manuscript entitled “Modifying the secretome of MSCs prolongs the regenerative treatment window for Encephalopathy of Prematurity”. The authors optimized the regenerative potential of MSCs by means of genetic engineering with transient hypersecretion of beneficial factors, IGF1, LIF, IL11 or IL10 which resulted in rescue of injured OLs and improved the functional outcome in EoP mice. However, there are still some points that need to be addressed by the authors to strengthen the manuscript.

 Major comments:

1.    I suggest changing the title to priming the secretome rather than modifying.

2.    In introduction, the authors hypothesized that reduced efficacy of postponed intranasal MSC treatment (in their previous study) could be the result of either impaired MSC homing or a limited regenerative potential of MSCs. However, other challenges are possible, for example encountering hostility within the transplanted microenvironment causing reduced engraftment time and lack differentiation and proliferative ability due to lengthy culture period.

3.    It is well known that MSCs secrete many factors and active molecules in their secretome even without modifications, how about the comparison of the secreted factors between modified and non-modified cells in the study. 

4.    EV-MSCs, does it mean Extracellular vesicles derived from MSCs? 

5.    Authors administrated the modified MSCs in disease mice model (EoP), did the authors investigate neural astrogliosis, apoptosis.  

Minor comments:

Please write the full name of the abbreviation when first stated.

Comments on the Quality of English Language

In the manuscript entitled “Modifying the secretome of MSCs prolongs the regenerative treatment window for Encephalopathy of Prematurity”. The authors optimized the regenerative potential of MSCs by means of genetic engineering with transient hypersecretion of beneficial factors, IGF1, LIF, IL11 or IL10 which resulted in rescue of injured OLs and improved the functional outcome in EoP mice. However, there are still some points that need to be addressed by the authors to strengthen the manuscript.

 Major comments:

1.    I suggest changing the title to priming the secretome rather than modifying.

2.    In introduction, the authors hypothesized that reduced efficacy of postponed intranasal MSC treatment (in their previous study) could be the result of either impaired MSC homing or a limited regenerative potential of MSCs. However, other challenges are possible, for example encountering hostility within the transplanted microenvironment causing reduced engraftment time and lack differentiation and proliferative ability due to lengthy culture period.

3.    It is well known that MSCs secrete many factors and active molecules in their secretome even without modifications, how about the comparison of the secreted factors between modified and non-modified cells in the study. 

4.    EV-MSCs, does it mean Extracellular vesicles derived from MSCs? 

5.    Authors administrated the modified MSCs in disease mice model (EoP), did the authors investigate neural astrogliosis, apoptosis.  

Minor comments:

Please write the full name of the abbreviation when first stated.

Round 2

Reviewer 1 Report

Comments and Suggestions for Authors

I have appreciated the efforts of the authors in revising the manuscript which is now acceptable for publication. I have only a minor point: the abbreviation for EV-MSC should be clarified in line 159, the fist time it is mentioned.